# The Other Side of the Perfect Cup: Coffee-Derived Non-Polyphenols and Their Roles in Mitigating Factors Affecting the Pathogenesis of Type 2 Diabetes

**DOI:** 10.3390/ijms25168966

**Published:** 2024-08-17

**Authors:** Alexis Ramerth, Brooke Chapple, Jeremiah Winter, William Moore

**Affiliations:** School of Health Sciences, Department of Biology and Chemistry, Liberty University, Lynchburg, VA 24515, USA; aramerth@liberty.edu (A.R.); bechapple@liberty.edu (B.C.); jwinter11@liberty.edu (J.W.)

**Keywords:** type 2 diabetes, coffee, alkaloids, caffeine, inflammation, oxidative stress, theobromine, theophylline

## Abstract

The global prevalence of type 2 diabetes (T2D) is 10.5% among adults in the age range of 20–79 years. The primary marker of T2D is persistent fasting hyperglycemia, resulting from insulin resistance and β-cell dysfunction. Multiple factors can promote the development of T2D, including obesity, inflammation, and oxidative stress. In contrast, dietary choices have been shown to prevent the onset of T2D. Oatmeal, lean proteins, fruits, and non-starchy vegetables have all been reported to decrease the likelihood of T2D onset. One of the most widely consumed beverages in the world, coffee, has also demonstrated an impressive ability to reduce T2D risk. Coffee contains a diverse array of bioactive molecules. The antidiabetic effects of coffee-derived polyphenols have been thoroughly described and recently reviewed; however, several non-polyphenolic molecules are less prominent but still elicit potent physiological actions. This review summarizes the effects of select coffee-derived non-polyphenols on various aspects of T2D pathogenesis.

## 1. Introduction

Roughly 10% of the United States population is currently living with diabetes, and T2D accounts for 90–95% of these cases [1]. The CDC reported that the number of adults diagnosed with diabetes has more than doubled in the last 20 years due to the rapid increase in obesity rate and decline in physical activity [1,2]. While type 1 diabetes remains the more prevalent among young adults and adolescents, the occurrence of T2D in individuals under 20 years of age has markedly increased over the last two decades, with 48,000 new diagnoses in 2021 [3,4].

Two major factors drive the progression of T2D: decreased insulin sensitivity in target tissues, primarily the liver, muscle, and adipose tissue; and, in the later stages of T2D, diminished insulin secretion from pancreatic β-cells [5,6]. The onset of insulin resistance is thought to be largely instigated by obesity, partially as a result of increased adipose tissue mass contributing to higher concentrations of circulating pro-inflammatory cytokines, non-esterified fatty acids, and glycerol [7]. Diminished insulin sensitivity results in decreased glucose disposal, thus triggering the release of excessive amounts of insulin to compensate for blunted glucose uptake [8]. Initially, this can help restore euglycemia, but continual exposure of pancreatic cells to excessive amounts of free fatty acids and glucose results in β-cell dysfunction and/or damage [9]. As β-cell function perpetually declines, blood glucose levels remain chronically elevated, resulting in clinically discernable T2D [10]. T2D progression can then lead to numerous complications, including macroangiopathy, cardiovascular disease, hearing loss, diabetic retinopathy, nephropathy, neuropathy, and damage to the lower extremities [11,12].

With the prevalence of T2D steeply increasing, the need for novel low-cost alternatives of effective treatment is obvious. Specifically, the use of natural products is appealing due to their widespread availability and affordability. The inverse association between coffee consumption and T2D risk has gathered an appreciable amount of evidence [13,14,15,16]. However, the exact substance or composite of substances responsible for this correlation remains unclear. Coffee consists of an abundance of bioactive molecules, including polyphenols, alkaloids, lipids, and polysaccharides. Polyphenols are well known to contribute to many of the bioactive properties of coffee—antioxidant, cardioprotective, anticancer, anti-inflammatory, and antimicrobial [17]. Furthermore, they were recently extensively reviewed in the context of T2D [18]. However, non-polyphenols also mediate numerous effects that either bolster the actions of the polyphenols or are solely responsible for some of the health benefits that are associated with coffee. The alkaloid class includes several compounds that have been shown to exert potent physiological effects, especially caffeine; trigonelline and its degradation product, *N*-methylpyridinium; theobromine; theophylline; harmane; and norharmane. Melanoidins are a subclass of polysaccharides, which are found in coffee and have also been shown to exhibit bioactive antidiabetic properties, specifically as it pertains to the prevention of T2D sequelae, including hypertension, cardiovascular disease, and various inflammatory diseases [19,20]. The purpose of this review is to provide an overview of the bioactive roles of prominent coffee-derived non-polyphenols in the context of T2D and its sequelae.

## 2. Components of T2D Pathogenesis That Are Affected by Coffee-Derived Non-Polyphenols

### 2.1. Inflammation and Obesity

Inflammation has been implicated in diabetes’s progression and may be at least partially responsible for the development of insulin resistance and β-cell dysfunction. A multitude of stressors, including obesity, can induce inflammation. As adipose tissue expands, adipocytes are crowded and starved of oxygen as their layers grow thicker, hindering blood-vessel infiltration [21,22]. Eventually, hypoxia and a lack of nutrients in adipocytes can lead to cell death [22]. Adipocyte death attracts macrophages, which are intended to remove the dead cells but simultaneously secrete proinflammatory cytokines and chemokines, such as tumor necrosis factor-α (TNF-α), monocyte chemoattractant protein-1 (MCP-1), IL-1β, and IL-6 [23,24].

Overproduction of TNF-α inhibits insulin-mediated glucose uptake by blocking the autophosphorylation of the insulin receptor [25]. TNF-α has also been shown to stimulate c-Jun N- terminal kinase (JNK) and the IKKβ/NF-κB pathway, which can increase IRS1 phosphorylation on serine/threonine residues, preventing the binding of IRS1 to the insulin receptor and tyrosine phosphorylation [26,27]. Thus, by these means, TNF-α accumulation impedes cellular insulin signaling and can eventually result in insulin resistance. The activation of IKKβ also causes the translocation of nuclear factor κB (NF-κB) to the nucleus by phosphorylating serine residues of the inhibitor IκB, leading to IκB ubiquitination and degradation [27,28]. NF-κB translocation promotes the transcription of additional pro-inflammatory mediators, including IL-1β, IL-6, IL-8, and more TNF-α, further exacerbating insulin resistance [27,29]. NF-κB activation also results in histone acetylation, an epigenetic modification that increases the transcription of a wide variety of genes, another means by which NF-κB upregulates the production of pro-inflammatory cytokines [30]. The transcription factor NF-κB is composed of two subunits, p65 and p50, which must join to allow NF-κB to carry out its function of binding to the DNA and regulating target gene transcription. NF-κB activity can increase when Akt phosphorylates IKKα on T23, causing IKKα and β to phosphorylate p65 at S534. All of these regulators are important in NF-κB pathway amplification in T2D, which plays a vital role in the development of insulin resistance by increasing proinflammatory cytokine production, which interferes with insulin signaling.

IL-8, which is upregulated by NF-κB, also furthers T2D pathogenesis by promoting insulin resistance in adipose tissue. IL-8 recruits macrophages to sites of inflammation, thereby adding to the existing proinflammatory cytokine/chemokine levels and inhibits insulin-induced Akt phosphorylation in adipocytes [31,32]. IL-8 also attracts neutrophils to the adipose tissue, which further amplifies insulin resistance [31]. Neutrophils secrete elastase, a proteinase, which can be taken up by adipocytes in the area and can cause the degradation of IRS-1 [33].

Another chemokine/cytokine which promotes insulin resistance is MCP-1, which was secreted by infiltrating macrophages in obese adipose tissue, as mentioned [34]. The adipocytes themselves also contribute to the rising MCP-1 levels. Hypoxia, experienced by adipocytes during obesity, elicits preferential MCP-1 secretion [35]. When adipocytes secrete MCP-1, it recruits additional macrophages and natural killer (NK) cells to the adipose tissue [35]. As macrophages accumulate, the levels of inflammatory molecules they produce also build, increasing TNF-α, MCP-1, IL-1β, and IL-6 [36]. The recruited NK cells also secrete IFN-γ and TNF-α, adding to the overall cytokine milieu in the adipose tissue [35]. Increased levels of these inflammatory cytokines have been linked to the development of insulin resistance. IL-6 upregulates the expression of suppressor of cytokine signaling 3 (SOCS-3), which interferes with the phosphorylation of both the insulin receptor and IRS-1 [37]. High levels of IFN-γ facilitate insulin resistance in the same way as IL-8, preventing Akt phosphorylation [38].

Another key inflammatory mediator involved in T2D pathogenesis is the NLRP3 (nucleotide-binding domain, leucine-rich-containing family, and pyrin domain-containing 3) inflammasome, a multimeric protein complex. The inflammasome is activated by IL-1β, and IL-1β levels are increased by both NF-κB pathway activation and macrophage infiltration, as previously mentioned. The inflammasome is primed for activation by the NF-κB and MAPK pathways [39], both of which upregulate NLRP3 transcription [39,40]. The inflammasome can then activate caspase-1, which cleaves precursor cytokines, including pro-IL-18 and pro-IL-1β, enhancing the existing IL-1β levels [41]. IL-1β promotes T2D pathogenesis principally by modulating β-cell mass and function [42]. More specifically, it causes Fas-triggered apoptosis, further NF-κB activation, and impairs insulin secretion. IL-1β influences NF-κB activation by initiating a signaling cascade, involving TNF receptor-associated factor 6 and the IKK complex, leading to NF-κB nuclear translocation. It mediates Fas-triggered apoptosis in pancreatic β-cells by stimulating iNOS, which generates NO [9]. β-cells already have levels of Fas ligand on their surface that are low enough to prevent susceptibility to apoptosis [43]. However, NO upregulates Fas expression, sensitizing the cell to apoptosis [43]. IL-1β-induced NO formation also disrupts β-cell function by causing a decrease in insulin secretion, as NO hinders the release of calcium from internal storage depots [44], which would otherwise promote cAMP accumulation and subsequent insulin secretion [45].

MAPK signaling comprises multiple cell-signaling pathways, including the extracellular signal-regulated kinase (ERK) pathway, the JNK pathway, and the p38 MAP kinase (MAPK) pathway [46]. These pathways can be initiated by proinflammatory cytokines, such as TNF-α [47]. Ligand binding engages the ERK, JNK, and p38 MAPK pathways, beginning with the activation of rat sarcoma (RAS) signaling molecules via the conversion of GTP to GDP with the assistance of the nucleotide exchange factor Son of Sevenless homolog 1 (SOS1) [46]. RAS, in its active form, binds to RAF proto-oncogene serine/threonine-protein kinase (rapidly accelerated fibrosarcoma, RAF), which can then activate the mitogen-activated protein kinase kinase kinases (MAPKKKs), mitogen-activated protein kinase kinases (MAPKKs), and the MAPKs in each pathway sequentially. This results in the eventual upregulation or downregulation of target genes, including NLRP3 [39]. The JNK, ERK, and p38 proteins, in their respective pathways, are the specific MAPKs, which are phosphorylated by the MAPKKs.

### 2.2. Oxidative Stress

Another well-studied instigator of T2D pathogenesis is oxidative stress, which can lead to pancreatic β-cell damage and impaired insulin secretion (Figure 1). The underlying cause behind oxidative stress is an imbalance in the normal equilibrium of pro-oxidants and antioxidants, resulting in an increase in pro-oxidant levels [48]. Pro-oxidants promote oxidative stress by increasing circulating levels of reactive oxygen species (ROS), molecules containing ROS, or interfering with antioxidant systems [49].

Multiple factors can cause this shift from pro-/antioxidant homeostasis, including hyperglycemia. Reduced insulin sensitivity in T2D, which leads to glucose accumulation in the blood, can result in increased glycation of proteins, nucleic acids, and lipids [50]. Catalase (CAT) is an enzyme that breaks down hydrogen peroxide, an ROS, into water and oxygen [51]. The glycation of CAT impairs its ability to function, decreasing its activity [52]. CAT levels are already naturally lower in β-cells compared to other tissues, and glycation can depreciate these levels further [53]. Decreased CAT activity raises ROS (hydrogen peroxide) levels in pancreatic β-cells, promoting autophagy and thereby lowering insulin secretion due to reduced β-cell mass [53] (Figure 1). ROS are capable of stimulating apoptosis by several mechanisms, including the activation of the JNK pathway [54]. Certain downstream targets of JNK are pro-apoptotic factors, which can cause the release of cytochrome C from the mitochondria into the cell to induce apoptosis [55]. These downstream targets include the pro-apoptotic proteins Bid and Bax [55].

Singlet oxygen is another common example of a destructive ROS [56]. It can mediate the oxidation of low-density lipoprotein (LDL), and oxidized LDL can, in turn, stimulate the degranulation of both neutrophils and eosinophils, allowing for the release of proinflammatory cytokines [57]. Proinflammatory cytokine release near adipose tissue can promote insulin resistance by the aforementioned mechanisms.

Another key antioxidant enzyme is superoxide dismutase (SOD), which facilitates the dismutation of superoxides into oxygen and hydrogen peroxide [58]. SOD can also be glycated in hyperglycemic conditions, hindering its activity [59]. Unregulated superoxide accumulation has been shown to activate uncoupling protein 2 in the mitochondria, allowing a greater number of protons to leak out of the intermembrane space [60]. Mitochondrial uncoupling diminishes ATP production efficiency and slows the rate at which ATP accumulates. Preventing an increase in the ATP/ADP ratio in the pancreatic β-cell interferes with potassium channel closure and therefore interferes with glucose-stimulated insulin secretion.

Glutathione is an antioxidant which aids in H_2_O_2_ neutralization [61]. Glutathione concentration decreases in T2D due to high oxidative stress and its increased activity in the polyol pathway [62]. As an antioxidant, a high percentage of glutathione is oxidized to glutathione disulfide by glutathione peroxidase [63]. After this reaction occurs, glutathione disulfide can then be recycled back to glutathione by glutathione reductase, with NADPH acting as a cofactor. However, if the capability to reduce glutathione disulfide back to glutathione is overwhelmed by a high burden of oxidative stress, glutathione disulfide is typically transported out of the cell or reacts with protein sulfhydryl groups in an attempt to restore the redox equilibrium, but depleting cellular glutathione levels in the process [64]. Thus, the increase in oxidative stress in T2D is linked with a decrease in reduced glutathione levels [65].

## 3. Purine Alkaloids

### 3.1. Overview

Alkaloids are a large group of nitrogenous organic compounds which are most commonly isolated from plants [66]. Over 20,000 different molecules make up the alkaloid class, some of which are present in food, as well as stimulant drugs, medicines, narcotics, and insecticides [67]. Well-known alkaloids include morphine, strychnine, quinine, ephedrine, caffeine, and nicotine [66]. Even in small doses, alkaloids exert strong biological effects on the body [67]. They have been shown to act as anesthetic, cardioprotective, anti-inflammatory, antibacterial, antitumor, antimitotic, and psychotropic agents [68]. Coffee plants possess three different types of alkaloids in large quantities: (1) purine alkaloids, such as caffeine; (2) pyridine alkaloids, such as trigonelline; and (3) β-carbolines, such as norharmane and harmane [69]. 

### 3.2. Caffeine and Its Metabolites

Caffeine is a purine alkaloid, which is subcategorized as a trimethylxanthine, a member of the methylxanthine class [70]. Its molecular structure contains two purine-like fused rings and three methyl groups located at positions 1, 3, and 7 [70]. Prominent sources of caffeine include coffee, tea, soda, and energy drinks [71]. An average 240 mL serving of brewed coffee contains about 97 mg of caffeine [72]. Caffeine concentration is affected by a wide variety of factors, such as the coffee species, method of preparation, brewing time, water temperature, water pressure, roasting, grind size, type of water, coffee/water ratio, and volume [18,73]. Of these, the most important determinants are the coffee species, method of preparation, coffee/water ratio, temperature, and pressure. A 7.5 g serving of coffee (95% Robusta + 5% Arabica) in a 25 mL volume of water, prepared with an espresso machine/portafilter at 92 °C and a pressure of 7 bar, produces a brew with 10.303 g/L of caffeine, which is one of the highest concentrations of caffeine found in a beverage.

Two prominent metabolites of caffeine are theobromine and theophylline, also classified as methylxanthines [74]. Theobromine is structurally similar to caffeine, with the exception of a methyl group missing from position 1, resulting in 3,7-dimethylxanthine [75]. It is found in chocolate, coffee, and tea [76]. Although coffee seems to contain only a small amount of theobromine, one study reported ~0.30 to 0.83 mg of theobromine per 100 g sample of coffee, and its content is especially relevant for coffee-based beverages containing cocoa, such as the café mocha [77], as there is 130 mg of theobromine in a 30 g serving of dark chocolate [76]. Theobromine content is known to be affected by fermentation and roasting, which demonstrates the importance of cultivation, processing, and preparing the beans for consumption in achieving a beverage with optimal health benefits. After harvesting, coffee cherries can be processed in several ways. Both natural and honeyed processing leave part or all of the pulp on the bean, which is left to dry for 2–5 weeks, allowing for relatively long periods of fermentation [18]. On the other hand, the washing method completely de-pulps the bean and allows it to ferment for a couple of days [18]. The fermentation process has been shown to reduce the concentration of theobromine due to biodegradation [78]. Interestingly, theobromine concentrations decline when coffee beans are roasted for 10 and 20 min but increase when roasted for 30 and 40 min [79]. This is probably because, over time, caffeine can break down into theobromine while roasting.

Theophylline is chemically known as 1,3-dimethylxanthine, lacking the methyl group at position 7 on caffeine [80]. It is present in tea, cocoa, and coffee and has been commonly used as a bronchodilator medication [30] to treat respiratory disease for more than 70 years [81]. Other noteworthy bioactive effects include CNS and cardiac stimulation, vasodilation, opposing inflammation, and acting as a diuretic [30]. Similar to theobromine, though trace amounts of theophylline are contained in coffee beverages—from ~0.79 to 0.95 mg per 100 g sample of coffee—coffee enriched with chocolate would supplement these levels [77]. Additionally, 4% of caffeine is converted into theophylline as it is metabolized [82]. The initial theophylline concentration in the coffee bean is decreased by roasting, steaming, and decaffeination [83]. Decaffeination and steaming have been shown to decrease the theophylline concentration to one half of the initial amount. 

### 3.3. Absorption and Metabolism of Caffeine and Its Metabolites

Caffeine is absorbed by the gastrointestinal tract within 45 min of ingestion [84]. It reaches its maximum concentration in the blood 15–120 min later, with 99% of the ingested caffeine being transferred into portal circulation [84]. Its polarity fosters its ready dispersal into the interstitial fluid [84]. As it is distributed throughout the body, it has an average V_d_ of 0.7 L/kg [84]. Caffeine is also lipophilic enough to cross all biological membranes [84]. Once it reaches the liver, it is further metabolized to produce dimethyl and monomethyluric acids, dimethyl and monomethylxanthines, and uracil derivatives [85]. Phase I (cytochrome P450 CYP) enzymes, predominantly CYP1A2, are the primary enzymes responsible for this processing, mainly found in the endoplasmic reticulum of the liver [74]. The major caffeine metabolite retained in the plasma is paraxanthine, whereas methylated xanthines and methyluric acids are the main metabolites excreted in the urine [74]. Caffeine can also be converted into theobromine through the action of CYP1A2 and theophylline by CYP2E1 [74]. Theobromine can be further metabolized into xanthine and then into methyluric acid, while about 12% remains as theobromine [85]. Theophylline is mainly converted to 1,3-dimethyluric acid, 1-methylxanthine, and 3-methylxanthine, with about 4% remaining as theophylline. 

### 3.4. Antidiabetic Effects of Caffeine

While coffee consumption has long been associated with a decreased risk for T2D, it is likely that caffeine is not the primary substance responsible for this effect. The available literature suggests that caffeine exhibits mixed effects. On one hand, it has been shown that caffeine can reduce excessive fat storage by increasing the metabolic rate, muscle performance, and lipolysis, all of which could promote overall glucose homeostasis. It has also been shown to attenuate inflammation and promote insulin secretion. On the other hand, caffeine appears to promote high glucose levels and to decrease insulin sensitivity in the short term [86].

In the context of inflammation, caffeine attenuates proinflammatory cytokine levels and acts as an adenosine receptor antagonist. Caffeine reduces proinflammatory cytokine levels by multiple means. It interferes with NF-κB binding to its target genes by inhibiting Akt-mediated p65 phosphorylation, which consequently blocks p50 and p65 translocation to the nucleus [87]. NF-κB inhibition subsequently impedes NLRP3 inflammasome activation, blocking its normal upregulation of proinflammatory cytokines, including IL-1β and IL-18 [88]. Caffeine also inhibits the activation of the MAPK pathway by interfering with JNK, ERK, and p38 activation [89]. It prevents the phosphorylation of these pathway intermediates in their respective pathways, interrupting the step in the MAPK pathways where the MAPKs (JNK, ERK, and p38) are phosphorylated by the MAPKKs. 

Adenosine receptor antagonism further contributes to the effects of caffeine as an anti-inflammatory agent, as it is a competitive antagonist of the A1 and A2 adenosine receptors (ARs) [90]. Through its operation as an A2AR antagonist, caffeine hinders NLRP3 inflammasome activation by an alternate mechanism: it blocks A2AR-induced NLRP3 inflammasome assembly [88]. The binding of caffeine stimulates a myriad of responses associated with adenosine receptor signaling, including the inhibition of neutrophil recruitment and activation, the prevention of macrophage cytokine secretion, and the hinderance of NK cell maturation and proliferation. Adenosine receptor antagonists have also been shown to negatively regulate proinflammatory cytokine secretion in peripheral immune cells, decreasing levels of cytokines like IL-6 and IL-1β [91]. Thus, caffeine may further attenuate inflammation in this way. 

Despite the plethora of evidence suggesting that caffeine promotes metabolic health, its short-term effects, less than a week of consumption, seem to stand in contrast to those data [13]. Several studies have shown that caffeine increases insulin resistance and promotes high blood glucose levels in the short term [13,14,15,16], with a two-fold increase in urinary caffeine and caffeine metabolites being positively associated with insulin resistance in nondiabetic adults [86]. This effect has been shown to hold when caffeinated coffee is consumed with a meal [19] meal, not 19. The effects were not seen with decaffeinated coffee [92]. Additionally, a recent review studied the effects of caffeine on blood glucose in randomized, controlled trials in diabetic patients. It was noted that five of the seven studies assert that caffeine intake results in a temporary impairment on the 2–3 h postprandial glucose response [93]. Though the function of caffeine as an AR antagonist is beneficial in the context of eliciting an anti-inflammatory/oxidative effect, this function is thought to be responsible for contributing to elevated blood glucose [94]. This is due to the fact that endogenous adenosine is a part of multiple metabolic signaling pathways, including those mediating glucose uptake, insulin secretion, glycogenolysis, and glycogenesis [95]. Blocking the AR has indeed been shown to impair glucose uptake in skeletal muscle [96]. The action of caffeine as an AR antagonist leads to the inhibition of Akt, as well as increased SREBP-1 expression, as the adenosine receptor normally promotes the phosphorylation of Akt and inhibits SREBP-1 expression [96]. Because Akt activation leads to the translocation of GLUT4 to the cell membrane, the inhibition of the adenosine receptor by caffeine also hinders glucose uptake [97]. Furthermore, SREBP-1 acts as a repressor of insulin receptor substrate 2, which normally acts as a molecular adaptor in the insulin signaling pathway [98]. Thus, the ability of caffeine to act as an AR antagonist also interferes with intracellular insulin signaling.

While a cursory look at these acute responses to caffeinated coffee suggests that caffeine intake may be harmful, other research indicates that caffeinated coffee might be beneficial. In a 24-week study, participants consuming four cups of caffeinated coffee per day saw no significant change in insulin sensitivity, fasting blood glucose levels, or biological indicators of insulin resistance [99]. This phenomenon is partially explained by the fact that caffeine habituation usually occurs after 1 week of regular coffee consumption [13]. Additionally, various experiments show that caffeinated coffee enhanced insulin secretion [64,65,66,67,68]. A 16-week randomized, controlled trial found that caffeinated coffee significantly decreased the area under the curve (AUC) of glucose and insulin tolerance tests by 21.5%, without achieving statistical significance [100]. Another study showed that insulin levels were significantly higher 30 min after caffeinated coffee consumption compared to decaffeinated coffee or water [101]. In a study of 20 healthy female participants who did not regularly drink coffee, caffeinated coffee consumption resulted in decreased blood glucose and cortisol levels [102]. A separate study showed that blood glucose levels did not change in ambulatory adults consuming caffeinated coffee compared to the control [103]. Whether habituation had been achieved prior to consumption in the latter study was not reported but is worth consideration in the attempt to characterize the rationale for these different effects. 

Taken together, caffeine appears to have some short-term effects that can superficially seem to be adverse. However, caffeine habituation can minimize the impact of these effects [13] to the extent that chronic caffeine consumption actually supports attenuated T2D pathogenesis by promoting insulin secretion, reducing fat stores, and decreasing inflammation. 

#### 3.4.1. Effects of Theobromine on T2D Pathogenesis

Theobromine, a caffeine metabolite, has also been shown to possess bioactive properties: acting as an anti-inflammatory molecule and insulin secretagogue.

Theobromine has been shown to reduce inflammation by preventing IL-1β-induced inflammatory responses and by acting as a phosphodiesterase inhibitor. Due to its structural similarity to caffeine, it is not surprising that theobromine also operates as a phosphodiesterase inhibitor [104]. This allows for decreased TNF-α expression and macrophage infiltration of white adipose tissue. Theobromine has also been shown to downregulate the expression of IL-1β-associated cyclooxygenase-2 (COX-2), prostaglandin E2 (PGE2), iNOS, NO, TNF-α, and MCP-1 production [105]. Theobromine inhibits MCP-1 secretion by hindering the differentiation of preadipocytes into mature adipocytes [106]. The mechanism is not yet confirmed, but it is possible that theobromine lowers the levels of the other inflammatory mediators by decreasing the activity of NF-kB. Theobromine either indirectly or directly blocks the phosphorylation and consequent degradation of IκBα [105]. The compound also depresses IL-1β levels [106], which might be mediated through NF-κB inhibition as well, since NF-κB regulates pro-IL-1β expression. Theobromine has also been shown to increase insulin levels in healthy men and women [107]; however, this mechanism remains uncharacterized.

As with caffeine, theobromine also appears to confer an acute effect on blood glucose that seems counterintuitive to studies suggesting its anti-T2D effects. Some experiments have observed an increase in blood glucose levels, in both healthy men and women, following theobromine consumption [107]. As another AR antagonist, theobromine likely increases circulating glucose levels in the same way as caffeine, inhibiting the metabolic pathways mediated by adenosine that are involved in glucose uptake [95,108].

#### 3.4.2. Effects of Theophylline on T2D Pathogenesis

A second metabolite of caffeine, theophylline, stands as a prominent bioactive molecule, as it has been shown to moderate inflammation, promote insulin secretion, and protect pancreatic β-cells. Theophylline reduces the expression of proinflammatory cytokines, including IL-8 and TNF-α, by acting as a histone deacetylase (HDAC) activator [109]. NF-κB upregulates the expression of certain pro-inflammatory genes, including IL-1β, IL-6, IL-8, and TNF-α, by promoting histone acetylation. However, HDAC downregulates those proinflammatory genes. Theophylline also inhibits matrix metalloproteinase-9 (MMP-9), which otherwise contributes to the activation of IL-8 and IL-1β [110,111,112]. 

Theophylline stimulates insulin secretion by acting as a phosphodiesterase inhibitor [113]. Phosphodiesterase (PDE) normally catalyzes the breakdown of cAMP, and its inactivation thus raises cAMP levels. An increase in cAMP concentration has been shown to enhance insulin secretion by affecting the ATP-sensitive K+ channel-dependent pathway of glucose signaling in rat pancreatic islets [114]. Finally, theophylline preserves the insulin content of pancreatic β-cells [115]. The mechanism has not been fully characterized, but theophylline was associated with a conservation of pancreatic β-cell mass in T1D biobreeding (BB) rats [115].

## 4. Pyridine Alkaloids

### 4.1. Overview

Trigonelline is derived from nicotinic acid and is chemically classified as a methylnicotinic acid [69]. Its structure contains a pyridine ring, along with a quaternary ammonium and carboxylate group, making this compound a zwitterion at pH = 8.3 [116]. Previous research has demonstrated that trigonelline has numerous beneficial effects, including acting as an antioxidative, antihyperglycemic, antihyperlipidemic, antihypercholesterolemic, anticariogenic, and antimicrobial agent [117]. Trigonelline is naturally found in coffee, coffee by-products, fenugreek seeds, and a variety of plants as a secondary metabolite [117]. Studies show that an average cup of American coffee includes a range of 34–60 mg trigonelline per 180 mL [118]. The most influential factors in determining final trigonelline concentration are the coffee species, roasting time, and roasting temperature [118]. Trigonelline content is inversely proportional to roasting time and temperature, as trigonelline degrades at temperatures above 180 °C [117]. It is also worth noting that Arabica coffee contains larger quantities of trigonelline compared to Robusta coffee [118].

Trigonelline thermal degradation results in multiple products, including *N*-methylpyridinium (NMP), an *N*-methylated derivative of pyridine [119]. NMP is categorized as a quaternary ammonium ion, a chemical attribute necessary for its many physiological functions [120]. It has been shown to oppose inflammation, oxidative stress, and fat accumulation. In dark roast coffee, NMP can reach concentrations of 785 μmol/L, which is sufficient to exert strong antioxidative effects [121]. Since its abundance is positively correlated with roasting temperature, NMP remains stable during numerous steps involved with coffee processing. A substantial amount of NMP is produced with temperatures above 220 °C; the ionic compound is present in both Arabica and Robusta coffee species; and it is stable in the midst of a variety of specialized treatments, like decaffeination and steaming, which is used to produce stomach-friendly coffees [122].

### 4.2. Absorption and Metabolism of Trigonelline and Its Degradation Product N-Methylpyridinium

Current research suggests that the absorption of trigonelline likely begins in the stomach, with the majority of absorption taking place in the small intestine [117]. Trigonelline begins to enter the plasma as early as 15 min after oral administration and reaches peak blood concentrations in 3–9 h [117]. Trigonelline has a half-life of roughly 5 h [123]. Appreciable amounts of trigonelline remain intact as it passes through the body, since 57.4% of trigonelline intake was shown to be excreted in the urine for males and 46.2% for females [123]. Trigonelline might be capable of crossing the blood–brain barrier, which has been demonstrated in transgenic mice [124]. Trigonelline metabolism is not well understood; however, four possible methylation and oxidation products have been identified in the plasma and urine: *N*-methylpyridinium, *N*-methylnicotinamide, *N*^1^-methyl-4-pyridone-5-carboxamide, and *N*^1^-methyl-2-pyridone-5-carboxylic acid [125,126,127]. Thus, further studies are warranted in order to fully characterize the pathways by which trigonelline is metabolized.

### 4.3. Antidiabetic Effects of Trigonelline

Trigonelline has been shown to elicit several antidiabetic effects, including improving β-cell function and insulin secretion, while also attenuating oxidative stress, hyperglycemia, insulin resistance, diabetic nephropathy, and diabetic hearing loss. It has been shown to restore pancreatic insulin levels to near-normal levels in insulin-resistant rats [128]. This is partially mediated by promoting glycogen synthase in both skeletal muscle and the liver [129]. It further promotes euglycemia by acting as an α-amylase and α-glucosidase inhibitor, controlling enzymes that hydrolyze carbohydrates to moderate postprandial glucose levels in the blood [130]. α-Amylase hydrolyzes complex starches to oligosaccharides, while α-glucosidase hydrolyses oligosaccharides, trisaccharides, and disaccharides to monosaccharides [130]. Trigonelline tends to promote the use of glucose for anaerobic ATP synthesis [131]. Though the underlying reason is not yet known, it is possible that trigonelline might promote glucose uptake specifically in type I muscle fibers, which are less oxidative than type II muscle.

Studies using the homeostatic assessment model for insulin resistance showed that trigonelline attenuates insulin resistance in STZ-induced diabetic rats being fed a high-fat diet [132]. Trigonelline also significantly increased GLUT4, peroxisome proliferator-activated receptor gamma (PPAR-γ), and Akt expression, as well as insulin receptor autophosphorylation in rats with T2D [133].

Trigonelline also appears to protect pancreatic β-cells by preventing β-cell apoptosis and ER stress. Research has also demonstrated that trigonelline prevents glucose-stimulated β-cell apoptosis by inhibiting caspase 3 in STZ-induced T1D rats, an effector caspase downstream of the extrinsic and intrinsic apoptosis pathways [134]. In addition, trigonelline protects β-cells from damage due to endoplasmic reticulum (ER) stress, as is evidenced by the downregulation of ER stress markers [135]. 

In the context of oxidative stress, research has shown that trigonelline increases SOD and CAT activity, concurrently with glutathione (GSH) levels [133]. Enhanced SOD and CAT activity, in turn, counteracts the imbalance of pro-oxidant/antioxidant homeostasis in oxidative stress. TNF-α can activate NADPH oxidase or overstimulate the mitochondrial electron transport chain, which can lead to excessive ROS (often superoxide) production as electrons escape from the chain and partially reduce oxygen [133]. Trigonelline decreases serum TNF-α levels by downregulating the expression of the tumor necrosis factor receptor superfamily 1a gene, thereby lowering ROS levels [133].

Trigonelline contributes to the reversal of diabetic nephropathy, as is indicated by its ability to significantly reduce blood urea nitrogen, creatinine, and albumin levels in STZ-induced neonatal diabetic rats [133]. Two more indicators of the alleviation of kidney damage are the improvement of tubular epithelial–mesenchymal transition and renal fibrosis [136]. These effects are mediated by upregulating Smad7 in proximal tubule epithelial cells [136]. Smad7 inhibits transforming growth factor-β, a key modulator of both renal fibrosis and epithelial–mesenchymal transition [137]. Trigonelline was also shown to inhibit apoptosis through the downregulation of p53, Bax, caspase 3, and caspase 9 in renal cells [138]. The ability of trigonelline to oppose the activation of the Wnt/β-catenin signaling pathway guards renal cells from further damage, as this signaling pathway promotes fibrosis in chronic kidney disease [133]. Trigonelline further inhibits fibrosis by restoring mesenchymal and epithelial proteins, as well as matrix metalloproteinase-9 to normal levels in renal cells [139]. 

In diabetic hearing loss, nerve growth factor levels are decreased, limiting its usual function of protecting cochlear structures [140]. However, trigonelline has been shown to augment nerve growth-factor levels, leading to the functional and structural recovery of hair cells [140]. In vitro research with spiral ganglion cells indicated that trigonelline caused a significant increase in nerve growth factor [140]. This mechanism has been fairly well characterized, as docking simulations have demonstrated that trigonelline specifically interacts with K88 and Y52 in the active site of the nerve growth-factor protein [140].

#### Antidiabetic Effects of NMP

NMP performs a variety of effects which oppose T2D pathogenesis. It has been shown to promote glucose uptake, oppose fat accumulation, and mitigate inflammation. At 0.09 mM, it was shown to promote glucose uptake in HepG2 cells by ~18.1% [131]. NMP was further shown to stimulate glucose utilization for ATP synthesis via aerobic respiration as opposed to anaerobic, as trigonelline does [131]. Thus, NMP facilitates the efficient use of the glucose taken in, with aerobic respiration producing a net of ~32 ATP molecules. 

PPAR-γ activation increases the expression and translocation of the glucose transporters GLUT1 and GLUT4 to the cell membrane, thus increasing glucose uptake in muscle cells and adipocytes, consequently reducing glucose plasma levels [141].

NMP-rich coffee was shown to significantly reduce body fat in both healthy and pre- obese subjects [104,126]. Although the mechanism is currently unknown, it has been suggested that NMP promotes the catabolism of stored fat [142]. Typically, during times of increased energy demand, whether due to exercise or fasting, lipolysis and beta oxidation are facilitated by several factors. Catecholamine release and subsequent interaction with beta-adrenergic receptors culminate in the cAMP-dependent activation of protein kinase A (PKA), which promotes the activity of adipocyte lipases. PDE is known to inhibit PKA activity by converting cAMP, an activator of PKA, to AMP. Cellular fatty acid uptake is then facilitated by one of several membrane-integrated transporters. Following activation by acyl-CoA synthetase, the fatty acid can enter the mitochondria via the carnitine shuttle system, which is regulated by the presence or absence of malonyl CoA. Malonyl CoA is converted to acetyl-CoA via malonyl CoA decarboxylase (MCD), which is activated by AMP-activated protein kinase (AMPK). Acyl- CoA carboxylase 2 (ACC2) counteracts the effects of MCD by converting acetyl-CoA back to malonyl-CoA, which prevents fatty acid entry into the mitochondria by blocking carnitine palmityl transferase 1. Thus, future research should investigate the effects of NMP on catecholamine production, PKA, PDE, MCD, ACC2, and AMPK activity, as well as on fatty acid uptake in general, to improve our understanding of the mechanism by which increased fat utilization is accomplished.

Pertaining to inflammation, NMP was shown to decrease the expression of MCP-1, C-X- C motif chemokine ligand (CXCL-10), and intercellular adhesion molecule (ICAM-1) in human adipocytes [143]. MCP-1 is known to recruit macrophages and natural killer (NK) cells to the adipose tissue, as previously mentioned. ICAM-1 mediates the infiltration of macrophages into the adipose tissue by allowing immune cells to attach to blood vessel endothelial cells, facilitating their movement to sites of inflammation [24]. CXCL10 seems to contribute to inflammation, leading to insulitis and destruction in pancreatic β-cells. CXCL10 induced sustained activation of Akt, JNK, and cleavage of p21-activated protein kinase 2 (PAK- 2), switching Akt signals from proliferation to apoptosis.

NMP was also shown to restore normal levels of adiponectin, thereby enhancing the effects of the hormone in controlling inflammation and improving insulin sensitivity [143]. Furthermore, NMP has been shown to downregulate TNF-α levels, reversing the suppression of Akt phosphorylation by TNF-α [143]. NMP seems to influence these processes by restoring PPARγ expression and inhibiting the activation of the pro-inflammatory mediator JNK [143].

### 4.4. Harmane and Norharmane

Besides caffeine and trigonelline, two alkaloids that are found in lower quantities in coffee are harmane and norharmane. These compounds belong to a group of alkaloids called β-carbolines, characterized by a tricyclic pyrido[3,4-b]indole ring structure [144]. Though research investigating the anti-T2D effects of these compounds is not as abundant as that of caffeine, several studies have shown that β-carbolines have a number of bioactive properties, including anticancer, antifungal, antimicrobial, anti-HIV, and antimalarial functions [145].

Norharmane is composed of a pyridine ring linked to an indole molecule. A derivative of norharmane, harmane, possesses the same structure but with a methyl group substituent at C-1. Besides coffee, other common foods and beverages containing harmane and norharmane include soy, maize, cookies, and fermented alcoholic beverages [146]. In coffee, the harmane and norharmane content of commercial blends ranges from 1.54 to 4.35 μg and from 4.08 to 10.37 μg, respectively [147]. This range is mostly due to coffee species, roasting time, and the volume of hot water used [147]. Robusta coffee tends to have higher amounts of harmane and norharmane, almost double that of Arabica [147]. Logically, hot water volume and harmane/norharmane concentration are positively correlated, as an espresso prepared with 70 mL of water contains more of the compounds than an espresso prepared with 20 mL of water. This is because, as the volume increases, more water passes through the coffee grounds, resulting in increased extraction of the compounds. Finally, though the difference is minimal, increased roasting times lead to decreased concentrations of harmane (by 3–4 μg), while norharmane is not significantly affected [147].

#### 4.4.1. Absorption and Metabolism of Harmane and Norharmane

As with trigonelline, the metabolic pathways of harmane and norharmane have not been fully characterized. After oral administration, at least part, if not the majority of harmane, is absorbed in the intestine [148]. The substance is thought to undergo first-pass metabolism, entering the liver immediately upon absorption [148]. In the liver, these β-carbolines are metabolized by the P450 enzymes 1A2 and 1A1, and minimally by P450 2D6, 2E1, and 2C19 [149]. Several metabolites result, among which are 6-hydroxy-β-carboline (a major product), 3- hydroxy-β-carboline, another hydroxy-β-carboline, and β-carboline-2-oxide [149]. After hepatic processing, the β-carbolines enter systemic circulation. It has been discovered that harmane achieves a 19.41% bioavailability within the circulatory system [150]. Twenty minutes after consumption, harmane concentrations reach a *C*_max_ of 9.588 nmol/mL [148]. Both harmane and norharmane are readily distributed throughout the body, with a V_d_ of 1.6 L/kg for harmane [148]. Excretion occurs primarily through the feces [150].

#### 4.4.2. Antidiabetic Effects of Harmane and Norharmane

Harmane and norharmane have both been shown to enhance insulin secretion [151]. Harmane and norharmane increase insulin secretion 2–3-fold in human Langerhans islet cell cultures, and harmane does so in a glucose-dependent manner [152]. It is likely that this effect is mediated by their action as an endogenous ligand to the imidazoline I3 receptor in pancreatic β-cells [152].

Harmane and norharmane also impede T2D progression through antioxidant and anti-inflammatory actions. Harmane has been shown to have a significant protective effect against the effects of H_2_O_2_ in yeast cells, operating as a scavenger of hydroxyl radicals (OH.) produced by the Haber–Weiss–Fenton reaction [153]. In general, β-carbolines have also been shown to quench superoxide, singlet oxygen, and hydroxyl radicals [154]. These antioxidant actions rely on the chemical structure of the compounds. Tryptophan and tryptamines, which are indole precursors of β-carbolines, also possess antioxidative properties, presumably because they can form a stable indole radical at the pyrrole ring after scavenging oxygen radicals [153]. The indole nucleus of β- carbolines is proposed to function in a similar manner [153]. β-carbolines also protect against oxidative stress by acting as monoamine oxidase (MAO) inhibitors [155]. They operate as specific, competitive ligands for the MAO enzymes, with norharmane inhibiting MAO-A and MAO-B and harmane displaying increased affinity for MAO-B [156]. The inhibition of MAO results in decreased levels of potentially toxic by-products—H_2_O_2_, aldehydes, and ammonia [155].

Harmane combats inflammation by inhibiting myeloperoxidase (MPO), a pro-inflammatory enzyme which plays a crucial role in the innate immune response and stands as a crucial marker for diseases associated with chronic inflammation [157]. For example, MPO is responsible for the eventual expression of surface adhesion molecules on endothelial cells, which, in turn, recruit leukocytes and elicit the production of cytokines and chemokines. Harmane significantly inhibited MPO at an IC50 of 0.72 μM [158]. Apart from inhibiting MPO itself, harmane was also capable of blocking the subsequent LDL oxidation carried out by MPO, with an IC50 of 0.52 μM [158].

Harmane has been shown to ameliorate the pathologically related issue of obesity, preventing lipid accumulation in 3T3-L1 cells [159]. Its mechanism of action relies on decreasing the expression of the adipogenic and lipogenic regulators CCAAT/enhancer-binding protein β (C/EBPβ), CCAAT/enhancer-binding protein α (C/EBPα), PPARγ, and adaptor protein complex 2 (AP2). The transcription factor C/EBPβ is upregulated early in adipogenesis to induce the expression of C/EBPα, which, in turn, activates the transcription of downstream target genes, including PPARγ and AP2. C/EBPα and PPARγ control the expression of fatty acid synthase, acetyl-CoA carboxylase 1, and stearoyl-CoA desaturase 1, all of which were downregulated in harmane-treated 3T3-L1 adipocytes [159]. Harmane has also been shown to induce adipocyte browning, causing a robust increase in UCP1 expression levels, the main protein responsible for non-shivering thermogenesis in brown adipose tissue [159]. This effect might also attenuate the onset of obesity. Harmane concurrently elicited increased levels of the mitochondrial transcription factors PGC-1α and PPARα [159]. PGC-1α and PPARα elevate heme oxygenase-1 expression, an Nrf2-regulated gene associated with the browning of perivascular adipose tissue [157,160].

## 5. Polysaccharides

### 5.1. Melanoidins

Melanoidins are brown polymeric compounds formed by the Maillard reaction, which occurs during coffee roasting [161]. Currently, there are two main types of melanoidins: polysaccharide and proteinaceous melanoidins, found in coffee and bakery items, respectively [162]. Coffee melanoidins primarily comprise polysaccharides, proteins, and phenols [162]. The melanoidin content of coffee brew has been reported to be about 1.7 to 4.3 g/L [163]. Concentration varies principally based on the roasting process [163]. With increased roasting intensity, melanoidin content increases, with strongly roasted coffees containing more fully formed melanoidins [161]. While melanoidin structure has not been fully elucidated, galactomannans and type II arabinogalactans are the most frequently observed polysaccharides used in their production [74]. Galactomannans possess a β-(1–4)-D-mannan backbone with α-D-galactose side groups [164]. Type II arabinogalactans consist of a β-1,3-galactan backbone and β-1,6-galactan side chains [165]. As melanoidins, arabinogalactans are often covalently linked to proteins, forming arabinogalactan–proteins [162]. Melanoidins can also be structurally linked to phenolic compounds, predominantly chlorogenic acids, which are polyphenols that are found in coffee [18].

### 5.2. Absorption and Metabolism of Melanoidins

Most melanoidins travel undigested through the superior part of the gastrointestinal tract [166]. They arrive at the colon intact, where they can be used as substrates for the gut microbiota. Before being metabolized by the resident microorganisms, melanoidins release the polyphenolic molecules, such as chlorogenic acid, that are often attached to them. Unprocessed melanoidins are excreted in the feces [166]. A portion of the melanoidins also enter the circulatory system— mostly melanoidins of a low molecular weight, with a small fraction of high-molecular-weight melanoidins [166]. The absorption of high-molecular-weight melanoidins occurs primarily in certain areas of the digestive tract: from the mouth to the small intestine and in the colon [166]. The primary site of absorption is across the intestinal wall, and the transporters which facilitate this are yet to be elucidated. Gastric conditions reduce melanoidin bioaccessibility by 2.7–4.3-fold, allowing these compounds to remain in the stomach, where they mediate key antioxidant and biological effects [166]. To date, there is limited research on melanoidin metabolism. However, a recent study provided some information on the metabolites produced by the microbiome [167]. Fermentative metabolism generated a high concentration of lactate and acetate short-chain fatty acids (SCFAs) accompanied by lower levels of butyrate and propionate. Lactate and acetate are the primary metabolic products of *Bifidobacterium* fermentation. Although yet to be confirmed, protein and carbohydrate derivatives are very likely by-products as well, due to their linkage to the melanoidins [168].

### 5.3. Antidiabetic Effects of Melanoidins

Melanoidins have been shown to convey several antidiabetic effects to coffee drinkers: regulating blood glucose and insulin spikes, acting as appetite suppressants, and as anti-inflammatory agents. One study explored how bread enriched with coffee melanoidins affected blood glucose and insulin levels, orexin-A and β-endorphin expression, and appetite [169]. With conventional bread consumption, the blood glucose levels of the volunteers reached their peak within 30–60 min, returning to normal at 180 min. Bread enriched with coffee melanoidins significantly lowered blood glucose at 60 min compared to conventional bread and restored insulin levels to pre-prandial levels faster than was observed in the control [169]. Some data suggest that melanoidins enhance the postprandial effect of ghrelin and oppose orexin-A and β-endorphin, which might serve to moderate glucose and insulin levels [169]. β-endorphin acts centrally to increase blood glucose by stimulating the sympathetic nervous system and the pituitary–adrenal axis [170]. Orexin-A may also raise blood glucose levels by sensing when blood glucose levels fall and modulating β-cell hormone secretion to maintain circulating glucose levels during fasting [171].

Melanoidins were also reported to reduce the appetites of human subjects, as well as daily energy intake by 26%. With obesity and over-nutrition being an important aspect of T2D pathogenesis, appetite suppressants can wholistically serve to prevent T2D onset. A recent systemic review found that galactomannans were the most effective dietary fiber for decreasing fasting blood glucose, HbA1c, LDL cholesterol, and triglyceride levels in people with T2D [172]. Another meta-analysis corroborated this finding, revealing that galactomannans could significantly reduce HbA1c and fasting blood glucose levels [173]. It was further shown that PAZ320, a galactomannan derivative, decreased postprandial blood glucose levels to a significant extent over the course of 2 h [174]. However, it should be noted that a small portion of their study population (5 out of 20) did not display this response [174].

Melanoidins also exert an anti-inflammatory effect, as they have been shown to reduce LPS-stimulated NO release in macrophages [19]. Melanoidins isolated from soluble instant coffee at 100 μg/mL were able to decrease NO release to 57% of initial levels in LPS-stimulated macrophage cells [19]. Past research has also illustrated the ability of roasted coffee products to downregulate NF-κB in LPS-stimulated RAW 264.7 cells, leading to decreased NO release [19]. Mice on a high-fat diet experienced a significant reduction in IL-1α, IL-1β, and TNF- α, by 31%, 15%, and 58% respectively, when given a daily melanoidin solution for 8 weeks [175]. This property of melanoidins is likely mediated by their release of phenols, which have been previously shown to reduce concentrations of TNF-α, NO, iNOS, IL-6, COX-2, IL-1β, and PGE2, among other inflammatory markers [176,177]. Phenols inhibit the phosphorylation of proteins within the NF-κB pathway and impede the synthesis and secretion of TNF-α, NO, COX-2, and PGE2 [176,178]. Additionally, the microbial metabolites, acetate and propionate, resulting from the processing of melanoidins, may contribute to the decrease in PGE2 levels, as both have been shown to lower PGE2 in astrocytes and F344 rats, respectively [179,180]. Melanoidins also regulate TNF-α and IL-1β levels by acting as matrix metalloprotease chelators, with the zinc ion located in their center, and by interacting with more specific structural features of the metallopeptidases [181]. Matrix metalloproteinase-7 and -12 cleave pro-TNF-α, allowing the release of active TNF from macrophages [182]. Another matrix metalloproteinase converts the inactive form of IL-1β into its active form [183].

## 6. Conclusions

Not only do the substances discussed in this article convey a variety of bioactive effects to coffee, a portion of their effects overlap (Figure 2), which arguably strengthens the overall magnitude and range of influence.

Caffeine, melanoidins, and theobromine all decrease NF-κB activity (Figure 3). Since NF-κB acts as a master regulator of inflammation, controlling the overactivation of this pathway reduces the progression of many different aspects of the proinflammatory state that often coincides with T2D. Caffeine prevents NF-κB from binding to its target genes by inhibiting Akt-mediated p65 phosphorylation, subsequently blocking p50 and p65 translocation to the nucleus. Theobromine either indirectly or directly interferes with the phosphorylation of and, thereby, the degradation of IκBα. Melanoidins release phenols, which have been shown to inhibit the phosphorylation of proteins within the NF-κB pathway, impeding the expression of TNF-α, NO, COX-2, and PGE2. Caffeine/theobromine, melanoidins, and trigonelline decrease IL-1β expression, but they accomplish this via different mechanisms, strengthening the overall degree and reach of IL-1β downregulation (Figure 3). Caffeine impedes NLRP3 inflammasome activation through its inhibition of NF-κB, hindering the inflammasome-mediated expression of IL-1β. Its function as an AR antagonist has also been shown to decrease IL-1β. Theobromine operates via the same mechanism of NF-κB inhibition. Melanoidins lower IL-1β expression by inhibiting matrix metalloproteinases and by releasing anti-inflammatory phenolic compounds. Trigonelline also inhibits matrix metalloproteinases.

Trigonelline, NMP, theobromine, theophylline, and melanoidins lower TNF-α levels (Figure 3). Trigonelline accomplishes this by decreasing the expression of *tnfrsf1a*. Melanoidins moderate TNF-α levels by two means—by being transporters of bioactive phenols and inhibitors of matrix metalloproteases. NMP has been shown to lower TNF-α levels by reversing the suppression of Akt phosphorylation by TNF-α. Theobromine seems to decrease TNF-α levels by hindering the activity of NF-kB. Theophylline reduces TNF-α expression by acting as an HDAC activator.

Controlling TNF-α by these compounds via different mechanisms allows for a more potent anti-inflammatory effect than either of the effects in isolation and, thus, may contribute to the epidemiological observations that coffee intake is associated with a lower risk of T2D development. Theobromine and NMP decrease MCP-1 expression (Figure 3). Theobromine does so by inhibiting the differentiation of preadipocytes into mature adipocytes [106]. NMP diminishes MCP-1 levels through its regulation of TNF-α.

Harmane, norharmane, caffeine, theophylline, trigonelline, and theobromine increase insulin secretion (Figure 4). Harmane and norharmane stimulate imidazoline I3 receptors in pancreatic β-cells, while caffeine enhances insulin secretion; however, the mechanism by which caffeine accomplishes this is unknown. The added effect of harmane and norharmane, which increase insulin secretion two- to threefold, enables coffee to have a robust effect on insulin secretion. Theobromine has also been shown to increase insulin levels in healthy men and women [99]; however, this mechanism remains uncharacterized. Trigonelline restores pancreatic insulin levels by promoting glycogen synthase in both skeletal muscle and the liver. Theophylline ellicits insulin secretion by acting as a phosphodiesterase inhibitor.

Harmane, caffeine, NMP, and melanoidins work to reverse excessive adipose tissue accumulation (Figure 4). Caffeine increases the metabolic rate by stimulating epinephrine secretion from the sympathetic nervous system. It also stimulates lipolysis through its action as a phosphodiesterase inhibitor. Harmane ultimately decreases the expression of the genes coding for fatty acid synthase, ACC1, and stearoyl-CoA desaturase 1. It also induces adipocyte browning. NMP significantly reduces body fat in healthy and pre-obese subjects, most likely by promoting the catabolism of stored fats. Melanoidins act as an appetite suppressant, preventing over-eating and subsequent fat storage. By these means, harmane, caffeine, NMP, and melanoidins might serve to prevent and/or attenuate adiposity and obesity, which is one of the main culprits responsible for T2D pathogenesis.

Trigonelline, NMP, and melanoidins work to decrease the chronically high blood glucose levels in the diabetic state (Figure 4). Melanoidins were associated with an increased postprandial ghrelin response and a decline in orexin-A and β-endorphin levels. Trigonelline stimulates glycogen synthesis, encouraging glucose storage. NMP promotes glucose uptake in HepG2 cells by an unknown mechanism. This decrease in glucose levels promoted by trigonelline, NMP, and melanoidins could attenuate T2D overall but also contribute to the prevention of several T2D sequelae, including diabetic neuropathy, retinopathy, and cardiovascular disease.

There are numerous bioactive effects of the non-polyphenols in coffee. On the one hand, this may imply that the consumption of the beverage might be superior, from an antidiabetic perspective, to using either of the compounds in isolation. It is possible that the epidemiological effects may be due to the overlapping bioactivities that are achieved by non-redundant mechanisms. However, the concentrations of the compounds used in most of the cited literature indicates that achieving a sufficient therapeutic blood concentration to elicit the observed effects of either compound in isolation would be out of reach by merely consuming the beverage. That, however, should not discount the benefits of coffee consumption or the pursuit of potent pharmacological interventions that might result from investigating the bioactive effects of these compounds individually.

## Figures and Tables

**Figure 1 ijms-25-08966-f001:**
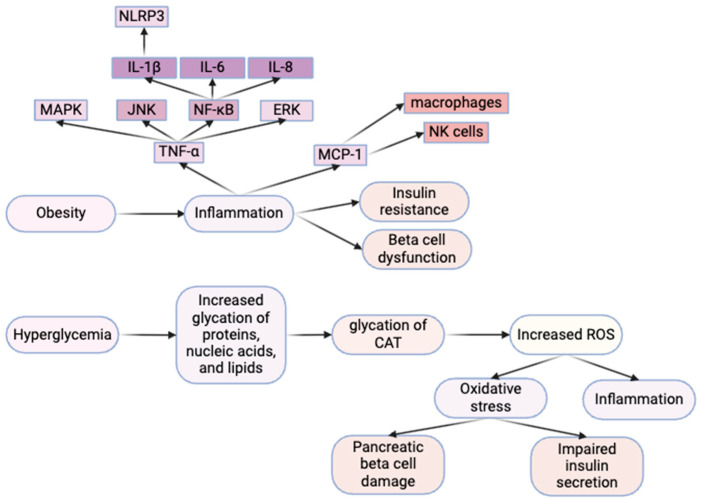
Connections between the causes of T2D, highlighting obesity, inflammation, oxidative stress and beta cell dysfunction. Image was generated by using BioRender.com.

**Figure 2 ijms-25-08966-f002:**
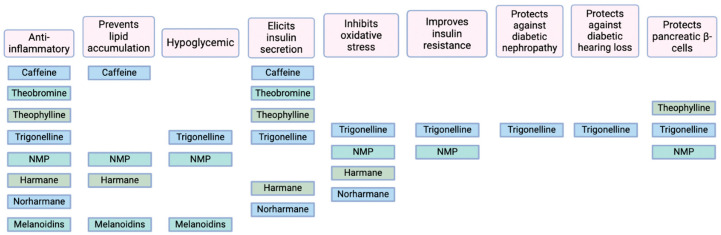
Summary of overlapping mechanisms through which each compound affects aspects of T2D and its sequalae.

**Figure 3 ijms-25-08966-f003:**
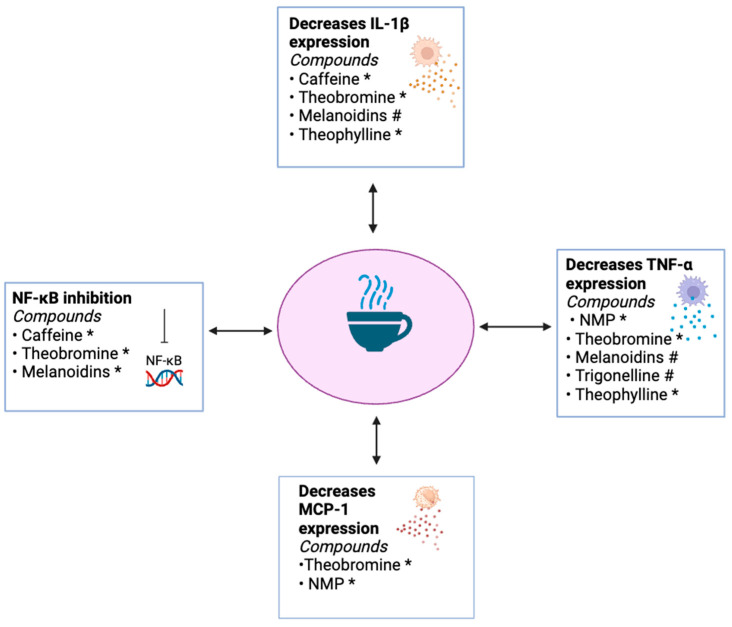
Mechanisms by which the reviewed compounds attenuate inflammation in T2D, focusing on the overlapping effects of NF-κB suppression, IL-1β secretion, TNF-α secretion, and MCP-1 secretion. * In vitro study. # In vivo study. Image was generated by using BioRender.com.

**Figure 4 ijms-25-08966-f004:**
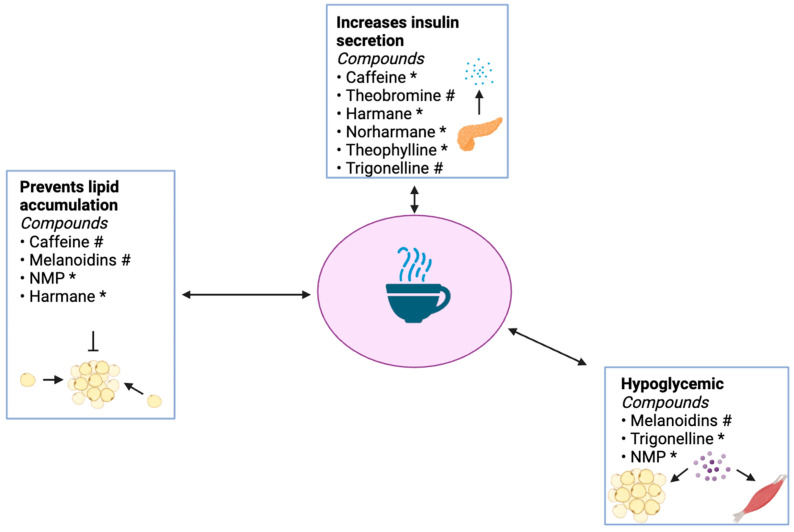
Corresponding mechanisms by which the reviewed compounds promote euglycemia in the adipose and skeletal muscle tissue. * In vitro study. # In vivo study. Image was generated by using BioRender.com.

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
