# Peer review of "The Other Side of the Perfect Cup: Coffee-Derived Non-Polyphenols and Their Roles in Mitigating Factors Affecting the Pathogenesis of Type 2 Diabetes"

_ijms, 2024, doi:10.3390/ijms25168966_

Round 1

Reviewer 1 Report

Comments and Suggestions for Authors

I have no substantive comments on the submitted manuscript, I think it interestingly describes the anti-diaberic potential of plant compounds contained in coffee, especially those other than caffeine. On the other hand, there are numerous editorial errors in the paper mainly missing spaces: vesres 32, 99, 119, 114, 152, 188, 193, 218, 221, 336, 347, 434, 515, 521,628, 644,649, 729, 733, 749, 752, 760)

Author Response

Comments 1: [I have no substantive comments on the submitted manuscript, I think it interestingly describes the anti-diabetic potential of plant compounds contained in coffee, especially those other than caffeine. On the other hand, there are numerous editorial errors in the paper mainly missing spaces: verses 32, 99, 119, 114, 152, 188, 193, 218, 221, 336, 347, 434, 515, 521,628, 644,649, 729, 733, 749, 752, 760)]

Response 1: [These were addressed throughout the document. Thank you for your feedback – it was very detailed and helpful.]

Reviewer 2 Report

Comments and Suggestions for Authors

In their review, the authors discuss several factors that can contribute to the development of type 2 diabetes, including obesity, inflammation, and oxidative stress. Coffee, a widely consumed beverage, has also shown a significant ability to reduce type 2 diabetes risk due to its diverse array of bioactive molecules. While the anti-diabetic effects of coffee-derived polyphenols are well-documented, this review focuses on the impact of less prominent but potent non-polyphenolic molecules found in coffee on various aspects of type 2 diabetes pathogenesis.

The review is well-structured and addresses a highly relevant topic. The subject matter is presented in an engaging and coherent manner. The organization and development of the content are logical, and the author communicates their ideas clearly and unambiguously, using appropriate terminology. The scientific work provides valuable clinical insights that can be applied to everyday practice. The entire paper is logically structured and statistically interpretable, with high-quality figures aiding data interpretation. The conclusions are thoroughly developed, offering sufficient depth and aligning the work with international research trends.

Comments:

·         It is recommended to correct typos and word separation errors, line spacing errors in the text and the table. There is no hyphen in the terms type 1 and type 2 diabetes.

·         In the case of figures, the numbering in the text boxes is irrelevant, it is recommended to remove them.

·         In the first half of the thesis, the connections between diabetes and inflammation, as well as oxidative stress, are discussed (Chapter 2). A better understanding and presentation of this part would be helped by the inclusion of a summary figure detailing the correlation of these factors with diabetes. In this way, the other figures in the thesis could be placed in a better context.

After completing the manuscript, I recommend this excellent work for publication.

Author Response

Comments 1: [It is recommended to correct typos and word separation errors, line spacing errors in the text and the table. There is no hyphen in the terms type 1 and type 2 diabetes.]

Response 1: [To the best of our knowledge, these have been fixed. The hyphens were removed from the text and title.]

Comments 2: [In the case of figures, the numbering in the text boxes is irrelevant, it is recommended to remove them.]

Response 2: [For the figures, the numbering in the text boxes was removed since the numbering conveyed no addition meaning to the figure.]

Comments 3: [In the first half of the thesis, the connections between diabetes and inflammation, as well as oxidative stress, are discussed (Chapter 2). A better understanding and presentation of this part would be helped by the inclusion of a summary figure detailing the correlation of these factors with diabetes. In this way, the other figures in the thesis could be placed in a better context.]

Response 3: [Figure 1 has been added to show the connections between inflammation, oxidative stress, and T2D and to highlight their progression.]

Reviewer 3 Report

Comments and Suggestions for Authors

The authors should introduce "empty spaces" between words in a high number of lines (l93, 144, 147, 152, 154 and ..)

The abbreviations should be introduced equally and after  ntroduction not be repeated once more (CAT).

The first three sections could be shortened because of the special issue of non-polyphenols.

The table in the conclusion (point 6) should be changed into a schema as in the other parts.

Catalase amount as an protective enzyme in the beta cells is also very low compared to hepatocytes or other cells.

Author Response

Comments 1: [The authors should introduce "empty spaces" between words in a high number of lines (l93, 144, 147, 152, 154 and ..)]

Response 1: [These have all been addressed to the best of our knowledge. We suspect that this happened as we were transferring the information from a regular Word document into the MDPI template. Thank you so much for making us aware of this issue.]

Comments 2: [The abbreviations should be introduced equally and after introduction not be repeated once more (CAT).]

Response 2: [The repeats of “catalase” and other compounds were removed and replaced with their respective abbreviations.]

Comments 3: [The first three sections could be shortened because of the special issue of non-polyphenols.]

Response 3: [Extraneous information has been removed from the first three sections, as the article is a special issue on non-polyphenols. A couple of sentences pertaining to polyphenols were kept since they connect this article to our recent publication describing the antidiabetic effects of polyphenols in coffee. This may help the reader get a more comprehensive picture of the issue.]

Comments 4: [The table in the conclusion (point 6) should be changed into a schema as in the other parts.]

Response 4: [We agree. This has been changed to show the information in a figure form and helps maintain consistency.]

Comments 5: [Catalase amount as an protective enzyme in the beta cells is also very low compared to hepatocytes or other cells.]

Response 5: [Correct, this caveat was added to the section explaining how there can be decreased levels of catalase in beta cells during T2D.]

Round 2

Reviewer 3 Report

Comments and Suggestions for Authors

Al points have been well performed